

# Biochemical profile in mixed martial arts athletes

Łukasz Marcin Tota[1] and Szczepan Stanisław Wiecha[2]

[1] Department of Physiology and Biochemistry, Faculty of Physical Education and Sport, University of Physical Education in Krakow, Krakow, Poland, Kraków, Polska

[2] Department of Physical Education and Health in Biala Podlaska, Jozef Pilsudski, University of Physical Education in Warsaw, Faculty in Biala Podlaska, Biala Podlaska, Poland, Biała Podlaska, Poland

## ABSTRACT

The study aimed to evaluate changes in selected biochemical indicators among mixed martial arts competitors in subsequent periods of the training cycle. The research involved 12 mixed martial arts athletes aged $25.8 \pm 4.2$ years competing in the intermediate category. Selected somatic indicators were measured twice. Biochemical indicators were assessed five times during the 14-week study period. Serum concentrations of testosterone, cortisol, uric acid, myoglobin, total protein, interleukin 6, and tumor necrosis factor, as well as creatine kinase activity were determined. One hour after sparring completion, there were significant increases in cortisol (by 54.9%), uric acid (22.0%), myoglobin (565.0%), and interleukin 6 (280.3%) as compared with the values before the simulated fight. The highest creatine kinase activity ($893.83 \pm 139.31$ U/l), as well as tumor necrosis factor ($3.93 \pm 0.71$ pg/ml) and testosterone ($5.83 \pm 0.81$ ng/ml) concentrations ($p = 0.00$) were recorded 24 hours after the simulation. Systematic observation of selected blood biochemical indicators in the training process periodization in mixed martial arts helps understand adaptive, compensatory, and regenerative mechanisms occurring in training athletes.

## INTRODUCTION

Mixed martial arts (MMA) is a sport growing in popularity around the world. The competitors must be comprehensively prepared in many aspects of physical fitness (*Chernozub et al., 2019*). Professional bouts consist of three or five 5-minute rounds, whilst amateur bouts are $3 \times 3$-minute rounds, with the winner determined by a referees' decision after the allotted time or the fight won ahead of time by knockout, technical knockout, submission, or disqualification (*Rainey, 2009*).

The implementation of training loads, regardless of the sports discipline, leads to metabolic disruptions, which are potent stimuli causing hormonal changes. Understanding the hormonal response to muscle damage and the direction of these changes will allow for a more precise training process control (*Philippou et al., 2017*).

Exploring and developing new training models is a priority for MMA coaches (*Kostikiadis et al., 2018*). To better understand the response to physical effort of an intensity similar to that in official sports competitions, simulated combat (sparring) is employed as part

Corresponding author
Łukasz Marcin Tota,
lukasztota@gmail.com

of the training process (*Amtmann & Berry, 2003*; *Kirk et al., 2021*). Previous studies with MMA athletes have confirmed acute and chronic changes that result from training load implementation or participation in competitions. However, these were mainly based on the anthropometric profile observation and on tests of physical fitness (*Tota et al., 2019a*) and salivary osmolality (*Vidal Andreato et al., 2014*)). A number of previous studies analyzed markers of muscle damage during simulated fights, but they failed to relate the magnitude of these indicators to the training period (*Ghoul et al., 2019*). Much valuable information on recording and evaluating biochemical indicators after competition and sparring is provided by *Coswig et al. (2016)*. A study by *Kirk et al. (2020)* is a very important source of data that allow to understand the specificity of the MMA sport and that emphasize the need for a close cooperation between the scientific community and sports practitioners. However, the latest research confirms the necessity for further extensive studies with MMA athletes, as the periodization of the training process is largely absent in this group (*Kirk et al., 2021*).

The applicability of simulated competitions in investigating biochemical responses in training periodization is confirmed by both theoreticians and practitioners of other sports, *e.g.*, judo (*Umeda et al., 2008*). The necessity to explore biochemical responses in MMA fighters arises from the high demands of the training. The implementation of training loads together with frequent aggressive strategies aimed at body mass reduction force coaches to employ a large array of biomarkers to streamline the training process (*Kasper et al., 2019*; *Kirk et al., 2021*). A better understanding of reaction to exercise helps avoid overtraining and optimize the training by recognizing athletes' individual adaptive and compensatory responses (*Chernozub et al., 2019*).

The study aim was to evaluate changes in selected biochemical indicators among MMA athletes during the preparation, regeneration, and competition periods.

## MATERIALS & METHODS

### Characteristics of the investigated competitors

The study involved 12 MMA athletes aged $25.8 \pm 4.2$ years with an average training experience of $11.8 \pm 2.6$ years. With regard to weight classes, there were four middleweight class fighters, two welterweight class fighters, four lightweight class fighters, and two bantamweight class fighters. They competed in the professional category (eight participants) and in the amateur category (four participants). The professional group represented the following organizations: Ultimate Fighting Championship, Pro MMA Challenge KSW, MMA Attack 3, Fight Exclusive Night, and FCB 9–Fight Club Berlin 9.

The research protocol was approved by the ethics committee of the regional medical chamber (approval No.: 7/KBL/OIL/2014). The athletes were informed on the study aims and course and provided their written consent to participate, in accordance with the recommendations of the Ethics Committee for Biomedical Research (*World Health Organization, 2000*). The subjects submitted valid results of their medical examinations.

## Study design

The observation period lasted 14 weeks and was conventionally divided into a preparation period (12 weeks) and a regeneration period (two weeks), after which the athletes took part in a simulated sports fight (competition period). The athletes declared an intensification of the applied training loads during the preparation period and a decrease of the volume and intensity of the training loads in the regeneration period. During the observation period, each athlete implemented eight training units per week (micro-cycle), each lasting $118.5 \pm 19.8$ min on average. Each micro-cycle included one day off and one day dedicated to wellness, on which the participants visited dry sauna. The average air temperature in the sauna was $95 \pm 5$ °C, with a relative humidity of 8–12%. The athletes reported three phases of warming that lasted $14 \pm 3.0$ min each and three phases of cooling. The simulated competition consisted of three rounds of 5 min each (with 60-second breaks between the rounds). In accordance with the current MMA rules, the period between the official weigh-in and the fight lasted 24 h. The average weight loss of the athletes during the analyzed period equaled $3.0 \pm 1.0$ kg ($1.1 \pm 1.3$ kg lean body mass and $1.9 \pm 0.8$ kg fat mass).

The athletes' nutrition or hydration were not interfered with throughout the study.

Anthropometric measurements were performed at the beginning of the preparation period (series I) and 24 h before the combat simulation (series III).

Biochemical indicators were assessed five times during the 14-week study period: series I—at the beginning of the preparation period; series II—after the 12-week preparation period; series III—after the two-week regeneration period, status before the fight simulation; series IV—1 h after the fight simulation; series V—24 h after the fight simulation (Fig. 1).

## Anthropometric evaluation

Body mass and body composition were indicated with a Jawon Medical (model IOI 353, Korea) body composition analyzer and the bioelectrical impedance method. Fat mass and lean body mass were established. Body height was determined to the nearest one mm with a Martin anthropometer (USA).

## Biochemical evaluation

On the day before the preparation period, after the 12-week training period, after the two-week regeneration period, as well as 1 and 24 h after the fight simulation, blood samples were collected from a cubital fossa vein by a diagnostician under certified laboratory conditions (PN-EN ISO 9001:2015), in accordance with the applicable standards. The blood was collected into Vacutainer EDTA tubes. Until centrifugation to separate the serum, the blood samples were stored in ice.

The immunoassay method (ELISA) and a DRG-type microplate reader (E-Liza Mat 3000, Medical Instruments GmbH, Germany) served to determine the following indicators in the serum: testosterone (EIA1559), cortisol (EIA1887), uric acid (AB83362), myoglobin (EIA2993), total protein (201-12-1151), creatine kinase (201-12-2091), interleukin 6 (IDEIH-1068), and tumor necrosis factor (IDEIH-1122).

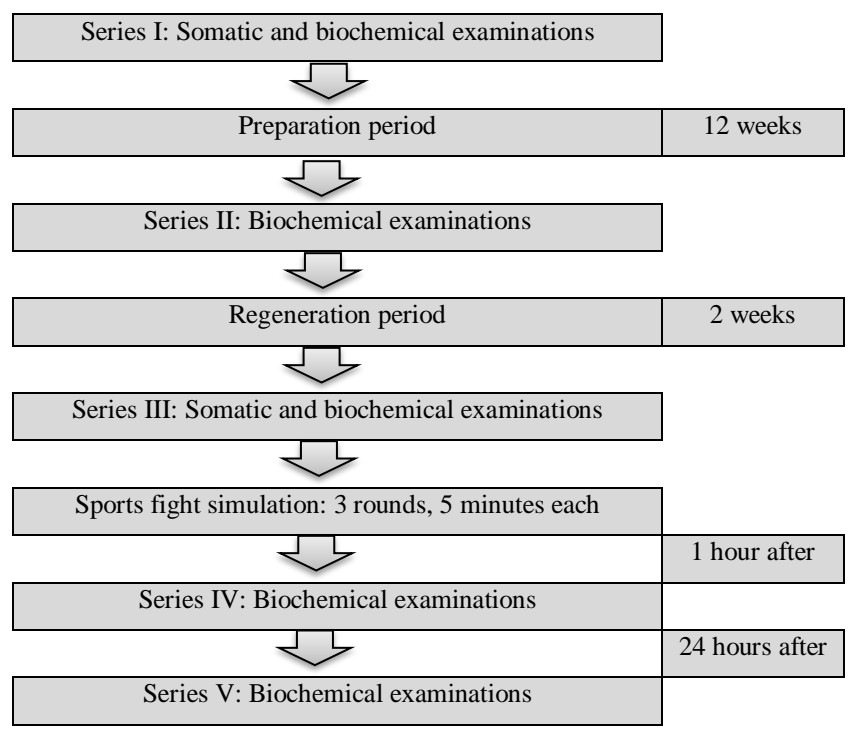

**Figure 1  Study design flowchart.**

The anabolic/catabolic balance indicator was established with the following formula: testosterone/cortisol ×100 (*Adlercreutz et al., 1986*).

Owing to the potential post-workout dehydration, all biochemical indicators were adjusted. The change in plasma volume (%ΔPV) was established with the formula by *Johansen et al. (1998)*. The *Kraemer & Brown (1986)* formula served to calculate the adjusted values.

All blood analyses were performed by qualified medical personnel.

## Competition procedure

The fights followed the MMA rules with respect to the venue and equipment requirements. The ring was surrounded by five ring ropes; its surface totaled 24 m$^2$. The fights involved professional refereeing. The competitors were divided into pairs, with the consideration of their weight category (difference not exceeding 5% of body mass) and technical and tactical skills. They performed three 5-minute bouts separated by 1-minute passive regeneration breaks.

## Statistical analysis and result presentation

The statistical analysis was performed with the R software, version 4.1.1. The Shapiro–Wilk test was used to verify the normality of the data distribution. To compare the five time intervals, a one-way repeated ANOVA test was applied or its nonparametric substitute (the Friedman test). We calculated the omega-square effect size ($\omega2$) to the ANOVA results

**Table 1  Changes in selected somatic indicators in the examined athletes.**

|  | BM [kg] | BH [cm] | LBM [kg] | FM [kg] | %F [%] |
|---|---|---|---|---|---|
| Series I | $79.5 \pm 9.0$ |  | $69.2 \pm 8.1$ | $10.3 \pm 1.4$ | $13.0 \pm 1.4$ |
| Series III | $76.5 \pm 9.0$ | $178.3 \pm 7.3$ | $68.1 \pm 8.6$ | $8.4 \pm 1.2$ | $11.0 \pm 1.5$ |
| $p$ | $\leq 0.01^*$ |  | 0.17 | $\leq 0.01^*$ | $\leq 0.01^*$ |

Notes.

BM, Body mass; BH, Body height; LBM, Lean body mass; FM, Fat mass; %F, Body fat percentage; Series I, Assessment before the preparation period; Series III, Assessment before the sports fight simulation explanations.

*Statistically significant $p \leq 0.05$.

for each study parameter. A pairwise $t$-test or pairwise Wilcoxon test (with Bonferroni corrections for multiple testing) served to calculate pairwise comparisons between time intervals.

# RESULTS

A significant ($p \leq 0.01$) decrease in body mass was recorded in the observation period between series I and III ($3.0 \pm 1.0$ kg: $1.1 \pm 1.3$ kg lean body mass and $1.9 \pm 0.8$ kg fat mass) (Table 1).

The Shapiro–Wilk test results for particular variables were as follows: $p = 0.103$ for cortisol, $p = 0.548$ for testosterone, $p = 0.379$ for testosterone/cortisol ratio, $p = 0.501$ for uric acid, $p = 0.001$ for creatine kinase, $p = 0.001$ for myoglobin, $p = 0.0189$ for interleukin 6, $p = 0.020$ for tumor necrosis factor, and $p = 0.032$ for total protein. Table 2 presents the mean values of the athletes' biochemical indicators throughout the observation period. At the beginning of the preparation period (series I) and prior to the commencement of the sparring matches (series III), all analyzed indicators fell within the reference values. One hour after the simulated competition, there were significant ($p \leq 0.01$) increases—and thus the highest concentrations in the analyzed period—in myoglobin ($122.23 \pm 57.11$ ng/ml) and interleukin 6 ($6.58 \pm 1.43$ pg/ml) as compared with the values before the simulated fight (series III). The highest creatine kinase activity ($893.83 \pm 139.31$ U/l) and tumor necrosis factor concentration ($3.93 \pm 0.71$ pg/ml) ($p \leq 0.01$) were recorded 24 h after the competition simulation. No significant changes were observed in cortisol concentration, testosterone concentration, testosterone/cortisol ratio, or uric acid concentration during the analyzed period (Table 2).

The Shapiro–Wilk test result for changes in plasma volume was $p = 0.002$. Plasma volume changed significantly ($p \leq 0.01$) between the subsequent study stages. The plasma volume changes recorded between the status before the fight simulation (the weigh-in was performed 24 h before the sports fight, in accordance with the MMA rules) and after the sparring sessions (series III–IV) equaled $-11.81 \pm 5.00\%$ on average. The plasma volume differences between the assessments obtained 1 h and 24 h after the competition amounted to $5.63 \pm 4.83\%$ (Table 3).

Tota and Wiecha (2022), *PeerJ*, DOI 10.7717/peerj.12708

**Table 2  Selected biochemical indicators in the mixed martial arts athletes in the observation period.**

| | Series I | Series II | Series III | Series IV | Series V | Effect size: omega² CI |
|---|---|---|---|---|---|---|
| Cortisol [ng/ml] | 168.71 ± 41.66 | 250.86 ± 45.98 | 179.77 ± 40.17 | 278.49 ± 34.42 | 208.43 ± 47.12 | |
| $p$ | 0.103 | | | | | 0.48 [0.26; 0.61] |
| Testosterone [ng/ml] | 4.97 ± 0.60 | 5.50 ± 0.70 | 5.74 ± 0.65 | 5.23 ± 0.46 | 5.83 ± 0.81 | |
| $p$ | 0.548 | | | | | 0.14 [0.00, 0.28] |
| Testosterone/ cortisol index | 3.11 ± 0.82 | 2.26 ± 0.45 | 3.36 ± 0.90 | 1.93 ± 0.28 | 2.18 ± 0.55 | |
| $p$ | 0.379 | | | | | 0.36 [0.13, 0.51] |
| Uric acid [μmol/l] | 300.69 ± 43.48 | 337.53 ± 44.30 | 297.86 ± 35.10 | 363.51 ± 53.22 | 302.99 ± 38.59 | |
| $p$ | 0.501 | | | | | 0.22 [0.02, 0.38] |
| Creatine kinase [U/l] | 106.51 ± 27.44 | 490.36 ± 133.60 | 123.12 ± 23.21 | 607.97 ± 90.96 | 893.83 ± 139.31 | |
| $p$ | Series I–II: $p \leq 0.01$*; series II–III: $p \leq 0.01$*; series III–IV: $p \leq 0.01$*; series IV–V: $p \leq 0.01$* | | | | | 0.90 [0.86, 0.93] |
| Myoglobin [ng/ml] | 16.03 ± 3.57 | 42.43 ± 18.78 | 18.38 ± 3.01 | 122.23 ± 57.11 | 33.43 ± 10.50 | |
| $p$ | Series I–II: $p \leq 0.01$*; series II–III: $p \leq 0.01$*; series III–IV: $p \leq 0.01$*; series IV–V: $p \leq 0.01$* | | | | | 0.66 [0.50, 0.76] |
| Interleukin 6 [pg/ml] | 1.57 ± 0.52 | 3.94 ± 1.34 | 1.73 ± 0.60 | 6.58 ± 1.43 | 3.01 ± 1.02 | |
| $p$ | Series I–II: $p \leq 0.01$*; series II–III: $p \leq 0.01$*; series III–IV: $p \leq 0.01$*; series IV–V: $p \leq 0.01$* | | | | | 0.75 [0.62, 0.82] |
| Tumor necrosis factor α [pg/ml] | 2.45 ± 0.45 | 2.64 ± 0.42 | 2.18 ± 0.35 | 2.84 ± 0.52 | 3.93 ± 0.71 | |
| $p$ | Series I–II: $p = 0.03$*; series II–III: $p = 0.04$*; series III–IV: $p = 0.03$*; series IV–V: $p = 0.02$* | | | | | 0.57 [0.28, 0.69] |
| Total protein [g/l] | 69.43 ± 3.66 | 70.56 ± 5.19 | 73.17 ± 4.12 | 83.14 ± 5.32 | 69.33 ± 3.68 | |
| $p$ | Series I–II: $p = 1.00$; series II–III: $p = 0.29$; series III–IV: $p \leq 0.01$*; series IV–V: $p \leq 0.01$* | | | | | 0.56 [0.37, 0.68] |

**Notes.**

Series I, Assessment before the preparation period; Series II, Assessment after the 12-week preparation period; Series III, Assessment after the two-week regeneration period, status before the fight; Series IV, Assessment 1 h after the fight; Series V, Assessment 24 h after the fight explanations.

*Statistically significant $p \leq 0.05$. Effect size: omega² CI - omega squared ($\omega2$) effect sizes to the ANOVA results for every parameter tested.

**Table 3   Plasma volume changes (% ΔPV) in the subsequent study stages.**

| Series | I–II | II–III | III–IV | III–V |
|---|---|---|---|---|
| %ΔPV | −1.29 ± 6.31 | −14.97 ± 6.35 | −11.81 ± 5.00 | 5.63 ± 4.83 |
| p | | I–II and II–III: $p \leq 0.01^*$; II–III and III–IV: $p \leq 0.01^*$; III–IV and IV–V: $p \leq 0.01^*$ | | |

**Notes.**

Series I, Assessment before the preparation period; Series II, Assessment after the 12-week preparation period; Series III, Assessment after the two-week regeneration period, status before the fight; Series IV, Assessment 1 h after the fight; Series V, Assessment 24 h after the fight explanations.

*Statistically significant $p \leq 0.05$.

## DISCUSSION

The study aimed to characterize MMA competitors' biochemical profile across the successive periods of the training cycle. To our knowledge, this is one of the few studies to employ such an extensive observation and analysis of biochemical indicators. It is our strong belief that the results of this study will contribute to a better understanding of training control in MMA fighters. The presented results should help better comprehend such training-related processes as fatigue, recovery, and adaptation. Moreover, our results confirm that MMA is a high-intensity sport and that the concentrations of biomarkers indicating the degree of muscle cell damage in athletes remain high even 24 h after the competition.

*Coswig, Ramos & Vecchio (2016)* compared muscle cell damage in MMA athletes after official and simulated competitions. They demonstrated that similar values of biochemical markers depicting the level of muscle damage could be observed after sparring and after competition. In a sports fight such as the ones in MMA during training and competition, some markers reflecting muscle damage may be related to the number of punches or kicks taken (*Wiechmann et al., 2016*). The literature repeatedly raises questions about the metabolic profile of MMA athletes. The complexity of training periodization stems simultaneously from the need to develop aerobic and anaerobic capacity and from the continuous training of skills, which include striking (punches, kicks, knee and elbow strikes), power attacks, and elbow attacks (*Kirk et al., 2020*). An essential element of sports training is the formation of grappling skills (punches, twists, throwing techniques using the legs) and submission on the ground (*Kirk, Hurst & Atkins, 2015*; *Tota et al., 2019a*).

In MMA, the phenomenon of body mass manipulation through rapid weight loss before fights followed by rapid weight gain is commonly encountered (*Kirk, Langan-Evans & Morton, 2020*). In isolated cases, it was also observed in the present study: the highest dehydration was recorded before the simulated fight (% ΔPV = −14.97). Numerous studies emphasize high adaptive capacity in elite athletes. It turns out that the reduction in blood and plasma volume due to loss of body mass prior to competition does not affect blood osmolality (*Yankanich et al., 1998*). As a limitation of the present study, the lack of assessing blood osmolality in the particular series should be pointed out, which precludes comparisons with the results of other authors. In the literature, one can find descriptions of planned and long-term (7-week) systematic weight losses (−18.1% body mass) (*Kasper et al., 2019*). However, the results of many studies confirm that acute weight cutting is dangerous and harmful to an athlete's health (*Matthews & Nicholas, 2017*). This

is mainly manifested in the inability to complete cardiac stress tests, endocrine disorders, hypernatremia, and kidney damage (*Kasper et al., 2019*). It is, though, worth emphasizing that the most recent research does not support the widespread belief of coaches and athletes concerning the need for rapid weight loss before the official weigh-in followed by rapid weight gain before the competition (*Kirk, Langan-Evans & Morton, 2020*).

Increased serum glucocorticoid hormone concentration in athletes indicates a normal response to stress stimulus (*Walker et al., 2017*). In the present study, the highest cortisol concentration (278.49 ± 34.42 ng/ml) was recorded 1 h after completing the fight simulation. No description of changes in this hormone during the preparation period was found in the literature, but the concentration measured after the 12-week preparation period was also high (250.86 ± 45.98 ng/ml). However, the changes in cortisol concentration during the analyzed period proved to be statistically insignificant. Different results were obtained by *Ghoul et al. (2019)* among 12 MMA athletes, who presented the highest cortisol concentrations directly after the simulated competition. *Lindsay et al. (2017)* demonstrated that immediate post-exercise immersion in cold water resulted in a faster decline in this hormone levels among MMA athletes. Therefore, further research should focus on accelerating regeneration processes, which can largely contribute to reducing the risk of injury in athletes.

Testosterone is involved in the restoration of damaged skeletal muscles, influencing the fusion of myoblasts forming multinucleated myotubes which give rise to muscle fibers. Myoblast fusion constitutes a key phase of muscle regeneration after physical effort (*Husak & Irschick, 2009*). Incorporating resistance/strength exercise into physical training increases testosterone concentration and satellite cell counts (*Mackey et al., 2007*). Monitoring changes in serum testosterone levels is often adopted among strength sports athletes. A decrease in the blood concentration of this hormone indicates processes of long-term adaptation to anaerobic power loads in athletes practicing fitness (*Chernozub, 2013*). In our study, the greatest decrease in testosterone concentration was reported 1 h after completing the fight simulation. However, it should be noted that throughout the observation period, changes in testosterone levels were not statistically significant. Changes in testosterone concentrations due to the implemented training loads are most likely related to the inhibition of the hypothalamic-pituitary-adrenal axis. Sustained inflammation and increased catabolism may inhibit this hormone secretion (*Kraemer et al., 2001*). An imbalance between physical training and rest periods results in reduced physical capacity, severe fatigue, and even overtraining (*Pankanin, 2018*). We did not observe a decrease in testosterone concentration resulting from dehydration before the official weigh-in (series III) as other authors have (*Coswig, Fukuda & Del Vecchio, 2015*). This is most likely due to a too little weight loss in our athletes (−3.9%) as compared with the one described elsewhere (−10.0% body mass) ((*Coswig, Fukuda & Del Vecchio, 2015*).

Training periodization involves evaluating testosterone and cortisol concentrations and determining their ratio (testosterone/cortisol). Decreases in this ratio occur in athletes practicing various disciplines both after competitions and after several weeks of training (*Marques et al., 2017*; *Tota et al., 2019b*). A simultaneous increase in cortisol concentration and decrease in testosterone concentration may further intensify catabolic processes at

the tissue level, thus reducing muscle strength and overall body capacity (*Barbas et al., 2011*). The lowest testosterone/cortisol ratio values observed in the present study occurred when measured in series II (after a 12-week preparation period), in series IV (1 h after the simulated fight), and in series V (24 h after the simulated fight).

Exercise-induced muscle cell damage triggers a repair response, macrophage entry into muscles, and an increase in interleukin 6. The changes in this cytokine concentration in the subsequent training periods in this study correspond with the observations by other authors (*Barbas et al., 2011*). A significant increase in interleukin 6 concentration (by 280.3%) 1 h after the competition indicated a rapid inflammatory response induced by high-intensity physical activity during sparring.

The high uric acid concentration after the simulated combat ($363.51 \pm 53.22$ µmol/l) is due to increased purine nucleotide degradation and damage to fast-twitch fibers under conditions of high energy demand. The elevated concentration of this marker may indicate a shift in the metabolic balance towards catabolic transformations accompanied by the breakdown of proteins and high-energy molecules and the release of free radicals. The high values reached after the preparation period ($337.53 \pm 44.30$ µmol/l) and after sparring may result from the high-intensity and high-volume exercise loads applied. They may also be due to insufficient hydration. The changes in uric acid concentration in the subsequent series of the study, confirmed by other authors (*Ghoul et al., 2019*), imply that high levels of anaerobic capacity are required from MMA competitors, which should be considered in training periodization.

High uric acid concentrations are accompanied by high creatine kinase activity. The highest increase in creatine kinase activity (by 625.2%) was observed 24 h after sparring compared with the value recorded before the competition. Reaching the peak level of this enzyme 24 h after the competition is in line with the findings by *Coswig, Ramos & Vecchio (2016)*. The high mean creatine kinase activity after the preparation period (490.36 U/l) and after sparring (607.97 U/l) may indicate muscle cell damage caused by intense physical effort—which, according to athletes, is encountered during sparring and wrestling sparring (rating of perceived exertion: $\geq 7$) (*Kirk et al., 2021*)—and by possible injuries induced by punching (*Cordeiro et al., 2007*). *Clarkson et al. (2006)* concluded that an increase in creatine kinase activity above 20,000 U/l after very intense physical effort did not require pharmacological treatment to prevent kidney damage. However, dehydration, a common condition in MMA, is a key factor increasing the risk of kidney damage by reducing myoglobin solubility (*Jetton et al., 2013*). The changes in myoglobin concentration during the observation period were most likely related to the increased metabolic stress resulting from the high-intensity physical effort implemented during the preparation and competition periods (*Ghoul et al., 2019*). As in the study by *Coswig et al. (2016)*, we believe that changes in myoglobin concentration can be a useful biomarker employed to streamline the training process in MMA.

## CONCLUSIONS

The biomarker changes (concerning myoglobin, tumor necrosis factor, creatine kinase, and interleukin 6) in training periodization allow to hypothesize that observing the

direction of these changes may prove crucial in monitoring compensation, regeneration, and supercompensation. In the context of muscle damage, these observations may enable a more precise control of the training process (*Philippou et al., 2017*). Nevertheless, further research is necessary to assess the impact of different forms of regeneration on recovery acceleration (*Lindsay et al., 2017*). The authors are aware of the study limitations, including no training load recording during the observation period and the focus only on the number and duration of the implemented training units. It is also worth noting that the weight loss in the experiment participants was certainly lower than that often seen in MMA athletes before their fights at combat sports shows. One can assume that the lack of rapid weight loss in our athletes did not cause a negative effect of dehydration on the investigated biochemical parameters, which was reported by other authors (*Coswig, Fukuda & Del Vecchio, 2015*). Therefore, the observed changes resulted from the implementation of training loads rather than acute weight cutting. Furthermore, technical or tactical actions during the simulation were not analyzed, unlike in some other studies (*Coswig, Ramos & Vecchio, 2016*). However, numerous such descriptions are found in the literature, in contrast to the coverage of changes in biochemical indicators during preparation, regeneration, and competition periods. This study aimed to investigate MMA athletes' biochemical profile in the preparation and competition periods. Observing changes in selected biomarkers helps diagnose the training loads in particular training cycle periods. The presented data confirm that MMA is a high-intensity combat sport and the concentrations of biomarkers reflecting the degree of muscle cell damage (myoglobin, tumor necrosis factor, creatine kinase, and interleukin 6) remain high even 24 h after the competition.

### Funding

This work was funded by Young researchers (No NN/602-147/14). The funders had no role in study design, data collection and analysis, decision to publish, or preparation of the manuscript.

### Grant Disclosures

The following grant information was disclosed by the authors:
Young researchers: No NN/602-147/14.

### Competing Interests

The authors declare there are no competing interests.

### Author Contributions

- Łukasz Marcin Tota conceived and designed the experiments, performed the experiments, prepared figures and/or tables, authored or reviewed drafts of the paper, and approved the final draft.
- Szczepan Stanisław Wiecha analyzed the data, prepared figures and/or tables, and approved the final draft.

## Human Ethics

The following information was supplied relating to ethical approvals (i.e., approving body and any reference numbers):

Bioethics Committee in Krakow

## Data Availability

The raw measurements are available in the Supplemental File.

## Supplemental Information

Supplemental information for this article can be found online at http://dx.doi.org/10.7717/peerj.12708#supplemental-information.

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
