# Peer review of "Biochemical profile in mixed martial arts athletes"

_PeerJ, doi:10.7717/peerj.12708_

## Round 0.1 · original submission · Major Revisions

As you will see from their detailed and thoughtful critiques, both reviewers recognised potential value in this study and have been positive in tone. However, both also recommend that significant re-analysis and reconsideration is performed. This includes a re-examination of the statistical tests (is the data normally distributed?) and inclusion of statistical analysis (in places where it is absent).

Some improved clarity in writing is requested by reviewer-1 and reviewer-2 has a list of suggestions which you should address.

Please detail all changes in a rebuttal letter should you decide to re-submit your work.

Reviewer 1 ·

Basic reporting

At points I feel the language used isn’t quite correct and an additional proof-read is required in order to provide clarity in a number of instances and would ensure the text could be understood more clearly by a wider audience. For instance, metabolic disorders are mentioned in the introduction yet it is unclear what is meant here or if ‘disorders’ is the correct terminology to be using. There are also a number of other instances where it seems the point is lost due to incorrect wording or phrasing being used. Some other examples are in lines 27, 40, 96 & 255.
The raw data was shared and the tables were generally good.

Experimental design

The general design of the study is good and the aims were relatively clear but some extra detail and specificity on the elements of the study would be beneficial. In the methods “intermediate category” is used to describe the participants, but it is unclear if this relates to the standard of the athletes or the weight category of the athletes. Standard MMA categories in terms of level would be amateur, semi-professional and professional and so if it relates to this it needs to be standardised to the generally accepted norms. It is also unclear if the athletes compete in the same weight category. The inclusion of ranges for things like body mass, lean mass, and fat mass would also be useful. The change in body mass in that time-period appears much lower than would be seen in MMA athletes, some explanation or discussion here would be beneficial to provide clarity. The lack of training load data does limit the interpretation of the data somewhat as well although if it were possible to include if every athlete completed all sessions and how the sessions were split e.g striking, grappling, conditioning etc this may help.

Validity of the findings

In terms of the statistical analysis, it is not made clear if the data were normality distributed or not. I also believe that the analysis used isn’t the most appropriate. A Wilcoxon test has been used but this compares each series in pairs as opposed to performing the comparisons altogether with post-hoc paired analysis subsequently used. A Friedman test may be more appropriate if data were not normally distributed, and a repeated measures ANOVA if the data were normally distributed.
There is also no inclusion of the data comparing series IV to V, unless there is a typo where it says III - V
In terms of your discussion and conclusion, there needed to be more discussion of the data from your study with a greater link to its significance/relevance in relation to the wider literature. The article would benefit by including the main findings from the present study within the first 1 or 2 paragraphs. You have also stated that this is the first study to explore this area and so it increases the reliability, which doesn’t automatically equate.

Additional comments

The authors are to be commended for implementing a time demanding and challenging study that aims to explore ‘real-world’ data from a sport that has seen significant growth in recent years. The data could provide some good insight into the metabolic responses to MMA training and competition and add to a limited depth of research in this area. However, whilst I see the value in the data presented, I feel there are a number of things that can be done to improve the relevance, impact and understanding of the data.

·

Basic reporting

Some important references are missing to provide context and comparison of the data.

Experimental design

Methods require details about any weight cutting that has taken place, methods used and magnitude of weight loss.
Reconsider the use of the Wilcoxon test for repeated measures data, as this may increase the error rate of multiple comparisons. Suggest replacing with Kruskall-Wallis test (if the data are non-parametric).

Validity of the findings

Data is provided and it clear, but please see previous comments regarding use of Wilcoxon for repeated measures and whether the results need to change accordingly.

Additional comments

To the authors,
Thank you for conducting an interesting and useful study in a group of athletes that is very difficult to conduct data collection with. I believe this will be a welcome addition to the literature, but there are some improvements required before publication. The main improvements is in the statistical analyses completed, consideration of other similar studies within the introduction and the discussion, and a deeper discussion of why your results occurred as they did and what they mean for applied practice. The authors also need to explain in detail the weight cutting practices employed prior to the simulated bouts and appropriately discuss these in relation to the biochemical results. Therefore I am recommending major revisions at this stage. I look forward to reviewing the resubmission and reading your work further.
Kind regards,
Christopher Kirk

Introduction

Line 34: Please state under which rule sets (professional/amateur) the length of rounds would differ.

Lines 37-38: I think this statement needs clarifying. Would metabolic ‘disruptions’ be more appropriate here than metabolic ‘disorders’ – as disorders suggests a chronic malfunction of the system, which I don’t think you’re implying here. Equally, are metabolic disruptions the only result of training loads and the only physiological aim of training? And are these changes positive or negative for the performance and health of the athlete?

Lines 41-42: I think a more appropriate reference may be used here. Suggest: https://www.ncbi.nlm.nih.gov/pmc/articles/PMC6090403/

Line 44: Suggest inclusion of this reference alongside Amtmann & Berry (2003) to provide a more recent evidence of how regular sparring is used in MMA training: https://journals.plos.org/plosone/article?id=10.1371/journal.pone.0251266

Lines 50-52: Suggest brief discussion of Coswig et al (2016) as a key paper on biochemical responses to sparring and competition: https://www.ncbi.nlm.nih.gov/pmc/articles/PMC5003304/. Also suggest consideration of https://pubmed.ncbi.nlm.nih.gov/28658085/ and https://www.redalyc.org/pdf/2730/273051167004.pdf as other papers to perform similar analyses (as summarised in: https://www.tandfonline.com/doi/pdf/10.1080/02640414.2020.1802093?needAccess=true). Whilst Ghoul et al is mentioned in the discussion, it would be worthwhile introducing it here to set up your data.

Lines 56-58: Suggest inclusion of https://journals.plos.org/plosone/article?id=10.1371/journal.pone.0251266 and/or https://pubmed.ncbi.nlm.nih.gov/29989458/ to demonstrate the combination and effects of training loads and weight cutting in MMA training.

Lines 61-62: Was a specific hypothesis formed for this study, or was it an observational study?

Methods

Line 67: Please define what is meant be ‘intermediate category’ here.

Lines 77-81: Was this program designed by the researchers? Or was this the participant’s regular training pattern? What did the preparation, regeneration and competition periods consist of? Please provide brief details of how each period differed from the others. What does the term ‘one day dedicated to wellness’ mean here?

Lines 81-84: Did the participant’s cut weight for the simulated bouts or did they take part at their habitual body mass? If they did engage in weight cutting, how much mass did they lose and over what time frame did this take place? Please consider what impact this might have on the biochemical data reported.

Lines 96-99: How were these measures controlled between trials?

Lines 121-122: Is a roped ring the ‘normal’ competition area for these athletes? Might this cause differences in technical/tactical occurrences in the bouts opposed to competing in a caged area?

Lines 129-130: Please clearly state which variables are somatic and which are biochemical. For the biochemical variables, is Wilcoxon the correct test to use? It seems the authors are comparing differences between more than 2 series at a time, which would require a repeated measures test (i.e. the Kruskall-Wallis test in the case of non-parametric data) to account for multiple comparisons errors.

Results

All statistical tests (including in text and tables): Please report p values ≤0.01 as such. P =0.00 suggests p is absolute zero, which is not possible. Please also provide the full Wilcoxon results and measures of effect size for all comparisons to allow comparisons between studies and inclusion in future meta analyses. Please see previous comment about whether the Wilcoxon test is the correct test to use for the biochemical variables.

Line 144: Please remove the term ‘highly significant’. P values are either significant or not. Please include effect sizes to demonstrate the magnitude of any differences.

Lines 154-159: Please reword this section to make it clear that 11.81±5% and 5.63±4.83% refers to plasma changes and not body mass changes. Please also see my previous comment about whether the participants engaged in weight cutting and to what extent during this study.

Discussion

Lines 164-167: Reconsider the use of the term ‘reliability’ here, as the number of results do not indicate reliability.

Lines 169-171: Suggest rewording of this statement as it’s difficult to understand what point the authors are trying to make. What is meant by the ‘right combination’? And why is it beneficial to obtain similar
biochemical values?

Lines 173-174: Please provide references to support this point – which studies raise this question?

Lines 179-180: Which data support this statement? Has any metabolic or energy data been collected from training or competition?

Lines 184-191: This is why the clarification of weight cutting magnitudes and practices is important for this study. Without this information we cannot know how or why these results occurred. Also, please support the phrase “numerous studies emphasize high adaptive capacity in elite athletes” with references and an explanation as to which adaptations are important (or possible) for weight cutting.

Lines 192-216: I think the authors need to present a deeper discussion about why the data may have occurred in this specific study and what this could mean for the training of MMA athletes. This section currently reads like a review of endocrine literature, rather than a discussion of findings.

Lines 240-241: Is this the highest individual increase? As the mean increase is not as great. Please reword to make this clearer.

Lines 243-245: Please describe what is meant by intense physical activity here and provide reference to https://journals.plos.org/plosone/article?id=10.1371/journal.pone.0251266 and/or https://sesnz.org.nz/wp-content/uploads/2020/09/Uddin-et-al-2020.pdf as the only studies that provide detailed MMA training load data.

Lines 248: Is dehydration common throughout the training period or only during the weight cutting period? If it’s not as common during the training period, then how likely an explanation for increased CK might this be?

Lines 255-257: What recommendations could be made from these data? When are the most physiologically intense periods? How might coaches manipulate these intensities and why?

---

## Round 0.2 · Minor Revisions

Please see the further minor suggestion of Reviewer-2. If you can add this data I will then recommend acceptance.

·

Basic reporting

Reporting is clear and in sufficient detail. Language used is professional and accurate, with good use of tables to display key results. The authors have addressed the key literature in this field in both the introduction and the discussion.

Experimental design

Research design is appropriate for the questions being asked whilst also avoiding unnecessary physical burden of the participants. Replication will be possible based on the details provided.

Validity of the findings

Data is robust, with sound statistical methods used. Conclusions are clear and reflect the data, which has been provided reviewed in it's raw format.

Additional comments

To the authors,
Thankyou for responding to the requested changes in a timely and thorough manner. I believe your study is now greatly improved and will provide a strong addition to the literature whilst giving researchers and practitioners a deeper understanding of the effects of an MMA training and competition period. The only minor amendment I request is the addition of omega squared (ω2) effect sizes to the ANOVA results for each test for use in future comparisons and meta analyses. I look forward to seeing this in print once this is completed (alongside any other changes requested by the other reviewer).
Kind regards,
Christopher Kirk

---

## Round 0.3 · accepted · Accept

Thanks for completing this - your attention to the revisions was appreciated and I am happy to recommend acceptance